# Understanding the Geography of Rape through the Integration of Data: Case Study of a Prolific, Mobile Serial Stranger Rapist Identified through Rape Kits

**DOI:** 10.3390/ijerph19116810

**Published:** 2022-06-02

**Authors:** Rachel E. Lovell, Danielle Sabo, Rachel Dissell

**Affiliations:** 1Criminology Research Center, Department of Criminology and Sociology, Cleveland State University, Cleveland, OH 44115, USA; 2Begun Center for Violence Prevention Research and Education, Jack, Joseph and Morton Mandel School of Applied Social Sciences, Case Western Reserve University, Cleveland, OH 44106, USA; dnb31@case.edu; 3Independent Researcher, Cleveland, OH 44115, USA; rldissell@gmail.com

**Keywords:** rape, sexual assault, rape kit, stranger, serial rapist, serial offender, environmental criminology, geography of rape, built environment, routine activity theory

## Abstract

Environmental criminological research on rape series is an understudied field due largely to deficiencies in official and publicly available data. Additionally, little is known about the spatial patterns of rapists with a large number of stranger rapes. With a unique integration and application of spatial, temporal, behavioral, forensic, investigative, and personal history data, we explore the geography of rape of a prolific, mobile serial stranger rapist identified through initiatives to address thousands of previously untested rape kits in two U.S. urban, neighboring jurisdictions. Rape kit data provide the opportunity for a more complete and comprehensive understanding of stranger rape series by linking crimes that likely never would have been linked if not for the DNA evidence. This study fills a knowledge gap by exploring the spatial offending patterns of extremely prolific serial stranger rapists. Through the lens of routine activities theory, we explore the motivated offender, the lack of capable guardianship (e.g., built environment), and the targeted victims. The findings have important implications for gaining practical and useful insight into rapists’ use of space and behavioral decision-making processes, effective public health interventions and prevention approaches, and urban planning strategies in communities subjected to repeat targeting by violent offenders.

## 1. Introduction

Routine activities theory provides an important framework for understanding how, why, where, and to whom sexual victimization occurs. As such, it is commonly applied within the victimology field. It posits that people are victims of crime when three conditions are met: a motivated offender with the ability to act upon inclinations, a suitable target, and the absence of a capable guardian who might deter the crime. This name stems from the idea that as a consequence of people going about their “routine” activities and social interactions, some individuals are more likely to be viewed as “suitable targets” [1].

Regarding sexually motivated crimes, routine activity theory is often applied in the context of higher education and/or college-aged victim-survivors—focusing on the role that alcohol and/or drug use played in the victimization [2,3]. This is largely because most are raped by someone known to them [4], implying that the victim-offender relationship plays a considerable role in the means of access to the victim and commission of the crime. However, for rapes where there is no victim-offender relationship—stranger rapes—the lack of capable guardianship hinges strongly on the built environment. This includes the physical structure of the domain (e.g., bridges, roads, parks, houses, transit systems, open space, infrastructure) and how the physical structure is being used and by whom as part of their routine activities.

Related to the routine activity theory, crime pattern theory posits that crime does not randomly occur across space and time but is patterned (as the name suggests). The observed pattern across space and time depends upon several factors, including the routine activities of the victim and the perpetrators and the perpetrators’ degree of familiarity with the locations—termed awareness spaces. These are areas or locations with which perpetrators are familiar and/or use regularly (e.g., near work, home, or recreation). Perpetrators are also aware of how their potential targets use these spaces. Thus, crimes are concentrated in locations that attract potential targets and provide opportunities to offend [5,6,7,8].

Given the predictability and patterned nature of the crime, research finds that stranger serial rapists tend to be fairly consistent across offenses, including their behavior and site selection [9,10]. For example, a recent study on the spatial proximity of rapes in a series for a large sample of serial rapists in the UK found that a majority of serial rapists committed rapes that were close in proximity (12 km). A small minority traveled much greater distances, crossing jurisdictional lines (100 km or more) in their sexual offenses. While important for understanding spatial patterns in rape series on average, the vast majority of the serial rapists in this study were linked to a small number of rapes (three or fewer rapes). Much less is known about the spatial offending patterns of more prolific rapists. Do different or more divergent patterns emerge?

Environmental criminological research on rape series is an understudied field compared to many other types of crime series. This is largely due to the deficiencies in the official and publicly available data of the potentially identifiable data about victims. Spatial data on rape series often do not contain two key elements necessary for a more complete understanding of the spatial patterns of rape—event and behavioral details of the already linked rapes (as discussed in [11]), especially for rapes where the offender is not officially linked to the series via the criminal justice system (as discussed in [12]). Concerning the first, to know how the environment patterns the crime requires information on not only the XY coordinates of the location of the rape but also what is at that XY coordinate, the nature of the victim-offender relationship, and the behavior of the victim and offender in the commission of the crime. For example, the rape of an intimate partner in a residence and the blitz rape by a stranger while the victim is walking or waiting outside requires the victim and the offender to use the environment differently. Most publicly available data only provide information on the offense’s location, date, and time. Concerning the second, most spatial research on rape series only includes data where suspects are officially linked by the criminal justice system via conviction, confession, or (more rarely) being a named suspect. What patterns emerge when examining a more complete series with more extensive details on the offenders, victims, and locations?

In this special issue on health geography and its implications for intervention, we conduct an extensive examination of a highly prolific, mobile stranger rapist (defined as someone completely unknown to the victim) linked by DNA to 22 rape kits through a novel application of spatial, temporal, behavioral, investigative, forensic, and personal history data. By applying more comprehensive data for a more complete series identified primarily through the testing of thousands of untested kits in the jurisdictions, we aim to gain a more thorough understanding of how this rapist used the built environment in the commission of his crimes. This case study has implications across several disciplines, such as victimology, law enforcement, spatial epidemiology, and public health in advancing our knowledge of the spatial patterns of prolific stranger rapists, more effective methods for interventions and prevention, and the impact that these crimes have on communities that are repeatedly targeted.

Our data are derived from an initiative to forensically test and follow up on the testing through investigating and prosecuting thousands of previously untested sexual assault kits in two neighboring U.S. urban jurisdictions. A sexual assault kit, also known as a rape kit, consists of items (e.g., fingernail scrapings, vaginal swabs) collected by medical professionals to preserve evidence from a victim of sexual assault. These initiatives are providing an immense amount of data on a large number of rapes collected over the span of approximately two decades but tested over a short period and include details on the rapes, suspects, and victims.

Data from untested sexual assault kits help fill previously mentioned deficiencies in our understanding of rape series by providing a more comprehensive accounting of rapes in a series, especially a stranger rape series. First, as strangers, most could not have been identified without the DNA testing, and the rapes would not have been linked (kit-to-kit matches)—masking the extent of serial sexual assault offending. Second, these data hinge on evidence gathered when the kit was collected, and not at conviction. Data from rape kit initiatives provide a more representative picture of a large number of reported sexual assaults, thereby providing a more comprehensive list of rapes in this offenders’ series.

In the following sections, we combine and summarize several different types of literature to more fully contextualize the study’s findings, including a background on the sexual assault kit initiative and how it is advancing our knowledge of sexual assault offending, sexual reoffending behavior, behavioral consistency in rape series, and spatial patterns in rape series. This is followed by a description of the data and findings, a summary of key findings, and a discussion of how this research informs our understanding of the stranger rape series and future policies and practices.

## 2. Relevant Literature

### 2.1. Sexual Assault Kit Initiative

Hundreds of thousands of sexual assault kits have languished for decades, untested in evidence storage facilities across the U.S. [12]. At times, these kits contain the only evidence linking a suspected perpetrator to the rape; however, this linkage requires the testing of DNA potentially contained within the kit. While these kits sat untested, many suspected rapists remained free and continued to rape [13]. Thus, untested kits contain potentially probative evidence of a crime and are a physical reminder of a failed criminal justice system response to rape [14,15]. The US Department of Justice’s Sexual Assault Kit Initiative was launched in 2015 to provide jurisdictions with funding to test and follow up on the testing of hundreds of thousands of previously untested sexual assault kits. 

The story of untested sexual assault kits is largely the story of marginalized rape victims. National estimates indicate that approximately 90% of all reported forcible sexual offenses victims are female [16]. Women and girls comprise the vast majority of those with untested kits (90 to 95%+), and of those with untested kits, the victims are disproportionately women and girls of color and/or marginalized populations [17,18].

The existence of large numbers of untested kits is attributed to several factors, including poor evidence tracking, outdated and ineffective investigative practices, lack of resources and personnel, crime lab case acceptance policies, victim-blaming beliefs and practices, and lack of understanding among law enforcement personnel about the neurobiology of trauma and the probative value in testing kits [14,19,20]. Traditionally, the utility of testing kits was seen primarily by law enforcement as a way to identify an unknown suspect when the victim-survivor was interested in prosecuting [19]. However, these kits provide much more to law enforcement. They can also connect offenders to previously unsolved crimes, confirm the identities of known offenders, possibly exonerate innocent suspects, and populate the federal DNA database [15].

### 2.2. Sexual Reoffending Behavior and Rape Series

Our understanding of crime is disproportionately limited to the crimes reported to law enforcement. This is even more relevant here since rape is the most underreported violent crime in the U.S. [21]. Nationally, 5 to 25% of all forcible rapes are reported to law enforcement [22,23], and approximately 5% or less lead to a conviction [22,24]. Thus, what we know about rapists is primarily based on those convicted of rape. Studies based solely on convicted offenders vastly underrepresent sexual assault offending and serial sexual assault offending. 

Criminal recidivism most often implies repeated adjudication in the criminal justice system [9], meaning that the offender is connected to two or more separate crimes via arrest and/or conviction [12], i.e., being “caught” once and then being “caught” at least one more time. The most recent national sexual recidivism rates find that approximately 8% of the released prisoners who had been convicted of a sexual offense were arrested for a subsequent sexually-based offense within those nine years [25]. Of note, this 8% sexual recidivism relies solely on official criminal justice system records—a completed prison sentence for a sexual offense followed by a subsequent arrest for another sexual offense within a limited time frame (nine years)—which, as discussed above, is very likely a substantial undercount of sexual reoffending.

#### Estimates of Serial Sexual Assault Offending

By examining rapists who confessed to undetected rapes (where the statutes of limitation had passed) [26], it is estimated that rapists commit an average of seven rapes per year per perpetrator with an average of almost four years of activity. This implies that rapists commit approximately 22 rapes from the time of the first rape and before conviction.

Data from SAK initiatives are starting to advance our knowledge of how common serial sexual assault offending is. Wayne County (Detroit, Michigan) and Cuyahoga County (Cleveland, Ohio)—two early adopter SAK initiative jurisdictions addressing a large number of untested kits—found that between a fourth and a third of the suspects connected to the now-tested kits can be identified as serial rapists [15,17]. While these numbers are much higher than the sexual recidivism estimates of 8%, they are likely still undercounts of serial sexual assault offending for several reasons. First, they only pertain to sexual assaults committed in the respective SAK initiative jurisdictions, not those in surrounding counties or other states. Second, using DNA from previously untested kits to establish serial sexual assault offending is predicated on a victim reporting the sexual assault, having a kit collected, law enforcement retaining the kit, and then law enforcement submitting the kit for testing. National estimates of how often victims who report a rape also have a kit collected do not exist; however, about half of all reported sexual assaults include a kit in one of the jurisdictions included in this study [27].

Furthermore, once submitted, suspects identified from testing the kit had to be connected to these newly tested kits either as a named suspect and/or through DNA. Two additional criteria were necessary for the kits to be linked if connected by DNA. The kit had to contain sufficient DNA from the suspect to meet eligibility for testing and meet even more stringent eligibility for the federal DNA database. A breakdown with any of these criteria would mean that the DNA would not make it into the database, and the rapes would not be linked. In other words, the rates of serial sexual assault offending are likely even higher than these a-fourth-to-a-third estimates. Data from rape kits and criminal justice records on linked rapists provide a more complete picture of the rape series.

### 2.3. Behavioral Consistency in Rape Series

Empirical evidence suggests that stranger serial rapists tend to have a great degree of behavioral consistency across offenses [9,28,29,30]. However, as these studies point out, much of the earlier literature on rape crime linkage (behavioral consistency and distinctiveness) was based on small samples of rapes where the offender’s *modus operandi* or MO was consistent and distinctive enough to be linked and result in a conviction (“solved”). More recent research has varied many of these conditions—solved vs. unsolved, use of DNA, larger samples, etc.—and finds that serial rapist behavior is consistent and distinctive enough to support the reliability of crime linkage practices [9,28,29,30].

However, data from previously untested kits suggest that once a more complete picture of a rapist’s sexual assault offending history is known, there is more versatility in sexual offending behaviors and victim preferences than once thought. An examination of serial sexual offenders identified via untested kits found that the rapists frequently rape both strangers and acquaintances and often exhibit intraserial variations in victim relationship, age, and even gender [12].

The intersections of offenders’ desirability, availability, and vulnerability help explain variations in an MO and their preferred victim selection process. Rapists are typically opportunistic—choosing the easier targets (as perceived by the rapist) with a greater chance of a successful outcome—i.e., raping and getting away with it [31]. Additionally, an MO is not “fixed” and may change for the rapist out of necessity to offend successfully. An MO involves the practical behaviors employed by the offender but is also an expression of their fantasy, which may change due to the offender’s internal affect, interests, and intentions. Therefore, an offender’s behavior—including MO—is complex and interactional, existing at the intersections of the offender’s internal drivers, life stressors, external opportunities, the victim’s behaviors and adaptations, and the environment they must navigate [32]. Lovell, Williamson, et al. [31] suggest that to be more effective in linking rapes, potential divergence from an MO and victim type preference must also be factored in. Thus, recent findings from previously untested rape kits might seem oppositional to the crime linkages rape literature but instead should be viewed as complementary, suggesting that rapes can be behaviorally linked but should cast a broader net than perhaps has been done in the past.

### 2.4. Spatial Patterns in Rape Series

Environmental criminology focuses on explicating how the environment affects crime patterns. Within this field, crime mapping blends applied criminal justice practices, geographical information systems, and science to understand the geographic nature of crime [33]. The theoretical frameworks often applied within these fields include routine activity theory and crime pattern theory. Within the routine activity theory framework [1], the built environment plays a key role in the “lack of capable guardianship” for certain types of sexual victimization—instances where the victims were raped by a stranger. Thus, individuals’ routine activities differ depending upon where they visit and how they use the built environment.

Crime pattern theory [34] holds that crimes tend to be patterned in and around the edges or boundaries of the nodes where people live, work, seek recreation, etc., and the paths that connect these nodes, thereby creating the perpetrator’s awareness space. Suitable targets who cross paths with offenders’ awareness spaces during their unaccounted-for time (when they are not expected to be at home, work, caring for others, etc.) are more likely to be victimized [34], in particular in areas where perpetrators believe there is a low probability of getting caught [35]. An expansion of crime pattern theory incorporates a temporal component, arguing that offenders’ use of the built environment inherently incorporates the time of day and season [36]. Overall, several factors impact rapists’ decision making regarding whom to target, including offenders’ and victims’ activities and where these activities occur, the environment, offender motivation, and situational cues [37].

#### 2.4.1. Journey to Crime and Mobility

The spatial and behavioral patterns of rapists vary based on the victim-offender relationship [12]. We focus here primarily on stranger rapists. The distance the offender travels to commit crimes is often referred to as the “journey to crime.” Stranger rapists tend to be more premeditated in the commission of their crime [37]. They often do not offend near their residences [38], although the victims of those rapes are often attacked near their residences [39,40]. Rapists who are more mobile have larger awareness spaces and, thus, more often use public transportation, are more familiar with multiple neighborhoods, more often commit property offenses during the sexual assault, and spend more time “roaming” [41].

One study found that the journey to crime for stranger rapists varied according to the offender’s motivation, categorized into four types [39]. Compensatory rapists are more impulsive, do not necessarily target victims, are less aggressive and brazen in the means of approach and the commission of the rape, and are generally unfamiliar with the crime location but offend closer to their residence. Opportunistic rapists are more familiar with the selected crime scenes, do not necessarily target certain victims, have less aggressive approaches, do not torture/terrorize (e.g., psychological, severe/extreme physical injuries to the victim, etc.), and travel the least amount of distance to commit the rapes. Angry/power rapists are more likely to specifically target victims, which may explain why they travel further distances to commit the rapes (e.g., need to travel further in search of specific victim attributes), are less familiar with the selected crime scene, are more aggressive in their approach (e.g., blitz) but do not torture/terrorize. Sadist rapists are the least common type of stranger rapist. They are less familiar with the selected crime scenes, do not target specific victims, and are not aggressive in their approach, but almost always torture/terrorize victims.

In terms of mobility and the consistency of site selection for serial stranger rapists, most function within limited environments [10,42], pattern themselves geographically, and often are relatively consistent in their site selection across offenses [10,39,43]. The specific characteristics of the crime sites do vary across sites [9], but the type of location selected (e.g., residence, neighborhood, shopping center) tends to stay consistent. However, most of this scholarship is based on stranger rapists with shorter series [10]. Those with more extensive (three or more) series show greater diversity, as offenders adapt and learn to be more “successful” in their attacks over time. After configuring a successful strategy and not being “caught,” more prolific serial rapists tend to become more consistent in their use of specific sites or specific types of sites [10], revisit sites from previous attacks [43], and over time, as they become more established in their abilities, become riskier in the locations where they encounter victims, such as in neighborhoods and the offender’s home [10].

#### 2.4.2. The Outdoor Built Environment

As an extension of “neighborhood”, stranger rapists often encounter and/or assault victims outdoors [44], defined as outside and/or public use spaces [40]. In outdoor rapes, various temporal and spatial properties influence sexual offenders’ approach to victims, including the likelihood of using coercion, physical force, and hiding during perpetration [45,46]. Accessibility, opportunity, and anonymity influence the geographic risk of sexual assault. High-risk areas include city centers with high concentrations of alcohol outlets [40,44], neighborhoods with high-residential population turnover and high rates of robberies [44], entertainment districts [47], and liquor-serving establishments [48]. In poor suburban areas, the risk of outdoor rape is higher near subway stations, near schools, with large female residential populations, and heightened fear of crime [44]. Relatedly, researchers examined outdoor rapes in Campinas, Brazil, and found an increased risk of rape in areas that had all three of these—bus stops, bars, and residences [49]. However, the type of sexual assault matters. One study found spatial differences in adult and child sexual assault locations and that specific micro-areas (street segments and intersections) in Austin, Texas produced most of the reported sexual offenses [11].

### 2.5. Significance and Aims of the Study

Through the lens of routine activities theory and with a unique integration and application of spatial, temporal, behavioral, forensic, investigative, and personal history data, we explore three research questions:(a)What do we know about the personal history of this (extremely) *motivated offender* who was able to act on his inclinations for nearly a decade before being detected?(b)What do we know about the *lack of capable guardianship* (which for these stranger rapes primarily involves geography and the built environment where the sexual assaults occurred) and the extent and consistency of mobility and type of site across the offenses?(c)What do we know about the *targeted victims* and how they were targeted, namely, the consistency of offending behaviors and victim preference?

With this unique combination of data, this study aims to fill a practical knowledge gap in the offending patterns of extremely prolific serial stranger rapists. Spatial and temporal data are important and often consistent features in stranger rapes but provide an incomplete picture without additional information. This study fills this gap by combining spatial and temporal data with forensic data (providing details on a more complete series via DNA), investigative data (providing details on law enforcement practices, hypotheses, and how the offender was eventually caught), behavioral data (providing details on the rapists and victims’ actions), and personal history data (providing details on what is known about the personal life, routine activities, awareness spaces of the offender).

The findings from this study have implications in several areas. First, by knowing more about the nature of rape locations, criminologists, spatial epidemiologists, and law enforcement gain insight into the offender’s site selection and behavioral decision-making processes. This speaks to the importance of the built environment in the commissions of these types of violent crimes. Second, victimologists and social service providers gain practical and useful insight for effective intervention and even prevention strategies—strategies that could be implemented at the micro-level and do not rely as heavily on victims to carry the majority of the burden to prevent their own victimization. Third, public health professionals gain insight into how public health intervention approaches [50] can be applied in communities subjected to repeat targeting or “hunting”. Exposure to violence (even if not directly exposed) has a significant and long-term impact on the health of communities, which are often disproportionately communities of color [51].

## 3. Materials and Methods

### 3.1. Description of the Data

The data are derived from two initiatives to follow up on the testing (via investigation and prosecution) of thousands of previously untested kits that span nearly two decades in neighboring jurisdictions. The initiative in Cuyahoga County (but almost exclusively within the city of Cleveland, Ohio) began in 2013 with the formation of the Cuyahoga County SAK Task Force and is led by the Cuyahoga County Prosecutor’s Office. Their initiative includes approximately 7000 untested kits (approximately 5000 kits that had never been submitted for testing and 2000 from Cleveland Police that had some prior forensic testing using outdated technologies and practices during the same time period [15]. This implies that in this jurisdiction, the untested kits in the initiative represent almost all the kits collected during the time period), primarily from 1993 through 2011. The initiative in Akron (limited to the city of Akron) began in 2018 with the formation of the Akron SAK initiative, is led by the Akron Police Department, and includes approximately 1800 never-tested kits primarily from 1993 through 2018. Cuyahoga and Summit Counties (Akron is the county seat) border each other. This offender was linked via DNA to 19 rapes in Cleveland and 3 rapes in Akron for a total of 22 rapes (encompassing the forensic data).

We quantitatively coded the case files for rape kits linked to this offender for this analysis. This includes information about the rapes, the offender, the victims, the kits, the investigations (past and current), and the prosecutions (past and present). We also had access to some investigative details about the crimes from the case files and from the detectives working the cases then and now (as part of the SAK initiatives). Combined, these sources encompass the spatial, temporal, behavioral, and investigative data.

The personal history data have been collected from the rapist’s criminal history, public newspaper stories on his crimes, and information collected by the third author when she was a reporter for the (Cleveland) Plain Dealer. Data on the offender’s criminal offenses are based on his statewide administrative criminal record that interfaces with out-of-state databases. We followed up with online adult county dockets to verify details. Given the high-profile nature locally of this offender, a great deal of information on this offender is already in the public realm. However, we have intentionally chosen not to use the perpetrator’s name in this study. The qualitative data were derived from converting PDFs of the incident reports to text files, then cleaned by hand to correct conversion errors as part of a quality control process.

### 3.2. Analytical Plan

To examine these sexual assaults, we qualitatively coded the police report narratives of the incident report to capture aspects of the rapes that were not collected as part of the close-ended quantitative variables and for validation of the quantitative findings. This step was also taken to get an initial understanding of the shared themes and patterns within the offender’s 22 assaults as his crimes were most often described as ‘random’ by law enforcement. Thematic coding was performed by hand in ATLAS.ti (ver. 9) and imported into a free word cloud software that allows for visual customization of the thematic coding. Word clouds are a straightforward way to represent textual data visually. According to researchers, these visual designs highlight more frequently used words, codes, or themes by allowing them to occupy more prominence (i.e., size) in the final representation [52]. Word clouds can be used for the preliminary analysis of text and validation of findings [52]. The larger or more prominent the coded word(s), the more frequently it occurred across the 22 rapes.

All 22 cases were geocoded using addresses or location descriptions within the police reports. The coordinates for each rape location were created using Google Earth Pro (ver. 7.3.4.8248). This process was undertaken to capture these locations with exact XY coordinates more accurately, and subsequent maps were created in ArcGIS Pro (ver. 2.7.0) utilizing a KML layer to the shapefile conversion process. All data have been presented so that victims could not be identified, especially with the spatial data. The exact locations of the rapes have been obscured.

## 4. Results

### 4.1. Motivated Offender

Figure 1 below presents the word cloud for the thematic coding of the 22 rapes. These findings highlight key themes and patterns in the offender’s behavior (death threat, vaginal rape, humiliation, suffocate or choke), and the environment (bus stop, alley), and how the investigation ended (NFIL or no further investigative leads) for these rapes.

#### 4.1.1. Daily Routine Activities

We begin by providing a personal history of this prolific rapist to provide context for his daily activities and use of space. He was 28 years old in his first known rape in the series in 1995. His last known rape in the series was in the summer of 2004. He was finally caught in connection to nine rapes in late 2004 (charged in 2005). From approximately 1996 through 2001, he was a probation officer in Lake County, which borders the east side of Cleveland. As an employee of the court, he would have a solid working knowledge of the criminal justice system and have frequent contact with those entangled in the criminal justice system. He was also a former employee and/or student at a university in downtown Cleveland. Most of the time, when he was committing these offenses, he lived in Akron (county seat of Summit County). To get from the county seat in Lake County, he would have had to travel through the east side of Cuyahoga County to get to Akron, Ohio—approximately a 55-mile trip one-way traveling south. Thus, like his rapes, he was mobile in his routine activities.

#### 4.1.2. Criminal History

Table 1 is a summary of this criminal history prior to being connected to these 22 rapes in 2004. In 2001, the offender was connected to several sexually and domestically related crimes. These crimes temporally coincide with his departure from being employed as a probation officer.

Of note here is that although this offender did have some criminal convictions, because they were misdemeanors, none would have qualified him for entry into the state DNA database. In 1997, Ohio passed legislation that mandated DNA be collected from those convicted of felonies and uploaded into state and national federal DNA databases. However, this jurisdiction [18,27], like many others around the U.S. [53], has identified issues with ensuring DNA from offenders with qualifying offenses is lawfully entered into state and federal DNA databases—termed lawfully “owed” DNA. Therefore, although some kits were tested in the early 2000s, they could not have been linked to him.

#### 4.1.3. Finally Caught

In an almost unbelieve turn of events, this offender was finally linked to nine of these rapes from an anonymous note and a newspaper clipping sent to the Cleveland detective working the cases. The tip mentioned the offender by name. While a suspect in several of these rape cases, he was finally swabbed for DNA in late 2004 in connection to a probation violation in the 2001 harassment and menacing charge.

In the mid-to-late 2000s, Cleveland Police started submitting some untested kits for forensic testing as part of a pilot kit DNA project—some 250 kits [54]. These are the nine initial ones to which the offender provided a hit. Once caught and swabbed, he was convicted and sentenced to prison for several of these rapes. However, at this time, Cleveland and Akron still had thousands of unsubmitted kits.

Then came the SAK initiative. Those thousands of kits were submitted by Cleveland and Akron Police for testing to the state crime lab (the same crime lab that tested his prior kits). While he was in prison, the number of rapes in his series grew—first in Cleveland and then in Akron (tested in the early-to-mid 2010s). Thus, the extent of his sexual assault offending was not known until all of the kits were tested, providing a more complete series. It is important to note that he was not convicted of all the rapes in Cleveland.

### 4.2. Lack of Capable Guardianship: Built Environment

#### 4.2.1. Description of the Geography

Cleveland, Ohio has strong roots in the manufacturing sector. Spatially, the city is primarily divided into east and west, with the Cuyahoga River designating east from west and Lake Erie to the north. There is limited movement and overlap between the east and west sides. The east side of Cleveland—sometimes referred to as the Cleveland Crescent— is the historically “redlined”, disinvested area of the city [55], overwhelmingly comprised of African American residents (90 to 95%), and has the highest poverty and crime rates [55] (as a reference, the city is approximately 50% African American [56]). The near west side of Cleveland, where the offender committed most of these assaults, is an area where several white non-native groups initially settled (e.g., Irish, Italian, Polish) and are now comprised of racially/ethnically diverse working-class residents. In general, this area in the late 1990s and early 2000s was an area of transition, with some pockets of gentrification [57]. Cleveland has the highest violent crime rate in Ohio [58].

Akron, Ohio is located in Summit County roughly 40 miles south of downtown Cleveland and is the fifth-largest city in the state of Ohio. Like Cleveland, Akron also has a strong manufacturing history—known as the “Rubber Capital of the World”, given Goodyear Tire Company’s location in the city. The city of Akron is divided into two parts by the Ohio and Erie Canal with downtown Akron being centered in the middle. Akron is also racially diverse. Almost 60% of the residents are white and about 30% are African American [59]. Similar to Cleveland, Akron typically ranks within the top 100 most dangerous city lists each year in terms of crime, drug use, and gang-related issues [60].

#### 4.2.2. The Offender’s Journey to Crime

During much of his crime spree, the offender was extremely mobile in his routine activities. His rapes span two counties. Figure 2, Figure 3, Figure 4, Figure 5, Figure 6 are the locations of the sexual assaults, indicating just how mobile he was. Figure 2 provides information on his awareness space, noting where he worked, lived, and the locations of the sexual assaults.

Figure 3 visualizes the locations of the rapes on the east side of Cleveland along with the time of the rapes and a description of the site (if in a building) or the victim’s activity when approached (if outdoors). (More details surrounding the rapes are provided in Table 2’s crime matrix). His first known rapes are on the east side of Cleveland—areas closer to Lake County, where he worked and potentially had a greater chance of being recognized.

Starting in 1998 and until the end of his series, the offender began to heavily target the west side of Cleveland, although not exclusively, as seen in Figure 4. He revisits Cleveland’s east side and even transports one victim from the west side to the east side. As mentioned above, many Cleveland residents in their daily lives do not travel from east to west or west to east, which makes his mobility even more noteworthy.

Figure 5 indicates the three rapes he is linked to in Akron, much closer to his residence. All three occurred in 2003. Shockingly, two of the rapes in Akron were within hours of each other—the ones closest in temporal proximity. Because he could not maintain an erection (the only mention of impotence in the series) during one of the rapes, he released the first victim and began searching for another. He mentions this to the second victim that day—saying that he needs to ejaculate and he then will release her. The two victims were actually in the hospital at the same time, getting their rape kits collected. The medical provider completing the forensic exam for both of the women observes that it appears that the same offender committed both the rapes.

#### 4.2.3. Strip Club Theory

Nothing in the case file definitively points to the offender’s motivation for selecting certain locations and the extent of his mobility. One investigative theory by detectives was that his crime selection sites were connected to strip clubs, given the area and the time of the assaults. Specifically, he would leave strip clubs in the early morning hours sexually aroused and go searching for potential victims. We explored the possibility of this theory by plotting the assault locations with known strip clubs that were open during this time. Figure 6 indicates that several of his rapes happened near strip clubs—alcohol-serving establishments open until early morning. In particular to the west side, three strip clubs were within two to three miles of 13 of the 22 sexual assault locations.

However, without information directly from the perpetrator (information that should be met with a measured amount of skepticism), it is impossible to know if the strip clubs directly influenced his location selection. They may be indirectly related. In other words, consistent with the literature, neighborhoods with strip clubs are also areas of increased activity, especially in the early morning hours, increased walking and waiting often near bus and train lines by women and girls (because of reduced access to a personal vehicle), increased drug use and/or sex work, or any number of temporary or permanent vulnerabilities exploited by these offenders.

There are three home invasion rapes, all in the spring of 2004—two are clustered near each other on the west side of Cleveland, and another on the east side of Cleveland near downtown. They are remarkably different from his other rapes. The victims did not have the same type of vulnerability (e.g., walking or waiting, sex work, drug use). Two occurred near midnight. One was in the early afternoon. More details on these rapes are discussed below.

#### 4.2.4. Patterns and Conclusions

Consistent with the literature, this offender appeared to favor a particular location. His west side rapes are within approximately two miles of each other, although they span eight years (almost the entire period of time in this series). Most occurred in the early morning hours or late morning. He has a cluster of 13 rapes within a few blocks of each other. As indicated in Figure 7, in an area north of the interstate, he even went as far as grabbing victims from the same street as before, taking them under the same bridge underpass, or moving them to this similar block to assault them in an area he knew well. As shown, victims in many of these are attacked while walking or waiting outside. This cluster is near a major local rail line (“light rail”) station, bus lines, industrial open space, and an interstate. We examined the day of the week (results not shown) but did not discern a specific pattern.

### 4.3. Suitable Targets: Offending Behaviors and Victim Selection

Table 2 and Figure 1 provide information on this series’ spatial, temporal, and behavioral patterns. Both illustrate several themes in his rapes. Overall, this rapist greatly varied his offending pattern over time. He frequently (but not consistently) attacked in the late night or early morning hours. In 14 of the 22, he approached more vulnerable victims while walking and/or waiting, frequently (but not consistently) in the early morning hours. Two teenage girls were attacked while walking to or leaving school. A few women willingly left the scene of first contact under the guise of obtaining drugs. Others he forced to leave the scene. This walking-waiting approach appears to be relatively common in this jurisdiction [61].

There is little consistency in his victim selection. The victims were from various racial/ethnic backgrounds, ages (teenage to 55 years old), and different professions (school teachers, sex workers, students studying on campus, a woman unlucky enough to run out of gas, etc.) (The latter two variables are not presented in the table to aid anonymization).

His behavior in the rapes varied. He sometimes used a vehicle (often not the same one). He sometimes raped in a vehicle. He sometimes threatened with a gun. He sometimes transported great distances. He sometimes stole money from the victims. He sometimes approached in a vehicle and other times on foot. In fact, without DNA, many of these rapes are so different from each other that they likely never would have been linked to him had it not been for the initiative.

His rapes tended to be clustered temporally (we redacted to aid anonymization). There are several clusters of rapes that occurred within several months of each other. Two Akron rapes were within hours of each other. Many of his rapes were brazen. In one home invasion rape, the victim’s family was upstairs. Two rapes involved girls on their way to or from school in the late morning.

Two rapes stand out as unique from the others—the campus rapes. Between 2000 and 2001, he worked at and/or attended classes at this college campus, implying that he would have been around thousands of potential victims each day and have familiarity with the buildings. In spring 2000, he followed an international student to the bathroom in the middle of the day and began to brutally rape her before she fought back and ran away. Then, in the summer of 2001, he followed a student into an empty classroom, turned off the lights, beat her, and raped her. These rapes were his most brazen in that they had the greatest potential to be interrupted, have eyewitnesses, and result in him getting caught. Spatially, these rapes also stand out, as they were on an urban university campus near downtown.

#### Patterns and Conclusions

Despite the mentioned inconsistencies, a few behavioral patterns are fairly evident in the table and word cloud. First, he only raped female strangers. Second, he was frequently gratuitously violent—exerting much more violence than instrumentally necessary to force compliance from the victim to complete the rape. He frequently used strangulation as a method to control the victims. The victims frequently describe him as a tall, large, strong man who exerted great physical control over them (sometimes described as smelling like he needed to shower). Third, his rapes appear to become more angry/sadistic over time, forcing victims to complete more degrading, violent acts (details reacted for anonymity). Fourth, while not exclusively his MO, he did appear toward the end of his series to substantially change his pattern by breaking into women’s homes and raping them. Finally, starting around 2001, he appears to become more forensically aware and attempts to conceal his identity. He consistently starts using a condom, mentions this forensic awareness to the victims, makes them shower, douche, and/or rinse their mouths, wears gloves, and consistently tries to prevent victims from visually identifying him. He spends hours with the women, moving them from room to room—much longer than needed to complete the rapes. In one rape, he tells the victim that he shaved his legs and chest not to leave any hair behind. He is prepared. In all three 2004 home invasion rapes, he expends a great deal of effort to dispose of forensic evidence, where he might have been provided more time to do so due to the home environment.

In many ways, the rapes that had a less distinctive MO, more spread out over space and time, and entailed more vulnerable victims were less likely to garner the attention of detectives. The campus rapes received a great deal of local press, as well as some of the later cluster of railroad track rapes on the west side of Cleveland. Without the benefit of DNA, police actively investigated many later rape cases through more traditional means. They asked victims to come to the police station to examine mug shots of convicted sexual offenders. However, in many of the later rapes, he attempted to hide his identity. Even if they could recognize him, we now know a mug shot did not exist for him during this time. New stories from the detectives working the case at the time mentioned that they thought they “had a serial offender on our hands” because of the clustering of rapes of vulnerable women with drug-dependency issues and/or sex workers near or around the railroad tracks. We now know via DNA that some of these rapes were connected to this offender. However, detectives believed that several other rapes with a similar offending pattern, location, description of the suspect, and victim type were potentially committed by this offender. All of the women were offering to sell sexual services when approached by the suspect. However, none of these included DNA. Unfortunately, while these were all similar, they were not necessarily distinctive enough to connect without DNA. This offender was never charged for any rapes in these jurisdictions which involved no DNA. DNA linked him to forensic evidence in a rape in Erie, PA, but the details of that crime and if it was the same or different from his indecent exposure conviction in Erie, PA are unknown because this case was sealed.

These findings speak to the power of DNA for understanding rapists’ behavior by connecting rapes that are different enough from each other to not be connected, and the flip side of that—rapes by different offenders that are not distinguishable enough in MO.

## 5. Discussion

This study aims to assess what can be learned by conducting a “deep dive” into the spatial patterns and sexual assault offending behavior of an extremely prolific, mobile stranger rapist who was able to complete the rapes of at least 22 women before getting caught. This study relies on a novel application of spatial, temporal, behavioral, forensic, investigative, and personal history data to address this aim. Through the combined use of these data points, particularly the forensic DNA that linked him to these rapes via rape kits, we can better understand the geography of rape. Using the routine activity theory framework, we explore more about the motivated offender, the suitable targets, and the lack of capable guardianship, which here primarily focuses on the geography of rape.

### 5.1. Theoretical

Overall, our study finds mixed support for consistency across the offenses. The offender committed rapes spanning large geographical areas—most often away from where he worked and lived (but not always). In support of crime pattern theory [34], he did not randomly select sites. He was more consistent in his site selection than his victim selection and other behavior-related characteristics. While he certainly was mobile, he also repeatedly returned to the same locations, as indicated in [43]. Thirteen of the offenses were within a two-mile radius on the west side. The east side locations were within three and a half miles of each other, and his Akron assaults were less than three miles of each other. Additionally, many of the assaults were separated by years but within a few blocks of each other. Even for a mobile rapist, spatial followed by temporal clustering were some of his most consistent “calling cards”, especially in Cleveland’s near west side.

Compared to the spatial and temporal data locations of the assaults, his behavioral patterns varied a great deal. In particular, from these data, we do not find support for a consistent motivation type across offenses [39]. For some of these rapes that are closer to his home, he appears to be motivated in a compensatory manner. He seems to be motivated by anger/power and even sadism for many others. Regarding how his motivation affected his journey to crime, there is no indication from the investigation as to why he selected the sites that he did, so we cannot speak directly to his level of familiarity with the locations. However, his repeat usage indicates a degree of familiarity.

### 5.2. Methodological

Across the series, it appears that the offender began by sexually assaulting the most vulnerable—sex workers with drug dependency issues—and progressed towards offending less marginalized and vulnerable women before getting caught. While this pattern is anecdotally assumed to be common, there is insufficient research to support this finding. More research on stranger rapists with larger series needs to be conducted to explore this observed pattern in greater detail. However, what we know now is that many of these rapes might have been prevented with a more effective criminal justice response to the rapes of the most vulnerable—including testing all the kits earlier and providing better and increased trauma-informed and victim-centered support for victims in the reporting, and more effective support in the kit collection and criminal justice processes.

Two of his rapes were connected as part of a project within the SAK initiative to review case files for rape kits that had already been tested with now-outdated forensic technologies, but that testing did not result in a DNA hit. This project aimed to see if the available forensic evidence in these previously tested kits could be fully leveraged using more current forensic technologies to identify unknown assailants. A prosecutor who was reviewing the case files happened to be reviewing two rapes in similar locations and a somewhat similar MO on the same day. She wondered if the same offender could have committed them. Her review of the case files suggested that additional testing could potentially produce a DNA hit. The state crime lab agreed to test additional items including the kits that had never been forensically tested, adding two more to this stranger rape series. These two rapes might have never been linked to him had she not identified the possible connection through geographic similarities during a case review and if additional forensic testing was not available.

## 6. Conclusions

In terms of better understanding the motivated offender, our findings suggest that when a more complete series is known, stranger rapists are not as consistent as we might have initially assumed. Without DNA, the assumed consistency might be more of a function of the crimes that are similar enough to be linked and leave out those that were different enough not to be linked. In this case, it is unlikely that his Akron rapes, campus rapes, and home invasion rapes would have been linked and linked to him. In line with others’ findings [10], thanks to the DNA, we have information on his later series, in which he appears to be more organized and riskier in the commission of his crimes. Research from SAK initiatives indicates that serial sexual assault offending is more prevalent than prior studies have suggested [14]. This is the most prolific offender identified as part of the initiatives in these jurisdictions—having 22 known rapes is certainly an outlier. However, this aligns with criminologists’ estimated 22 rapes before being “caught” [62].

In terms of better understanding the targeted victims, we found substantial inconsistency with regard to the victims’ characteristics (e.g., race, age, profession, etc.). The only consistent demographic victim characteristics were that they were all female and all strangers. This finding suggests that for law enforcement seeking to link cases together without the availability of DNA, while not always consistent, geographic site selection might be more consistent across offenses than behavioral data. As discussed, it is very difficult to explain a rapist’s observed behavior because of the numerous interacting factors that come into play in the commission of the rape [32]. Seeking this knowledge directly from the offender is potentially helpful. For example, did he begin doing home invasions due to his increased forensic awareness? If yes, how and from where did this forensic awareness come? The campus rapes were his most different and brazen—why did he choose such a public location? What happened to him in his personal life in and around 2001 when he was convicted of several crimes?

In terms of better understanding the lack of capable guardianship, since our data combines spatial and temporal data with information on this offender’s personal history, forensics, and the event details of the rapes, we can understand his offending series. For example, while he seemed to prefer some locations, how he used the built environment in those same locations during the rapes changed. For the rapes where victims were walking or waiting when approached, he interacted much more with the outside physical environment. For the home invasion rapes, he interacted less with the physical environment. He had less public exposure, prolonging the duration of the rapes and likely contributing to him being able to dispose of some of the forensic evidence. These details would not have been possible to discern if only the locations of the rapes were known.

This case study provides a model for incorporating and applying different data points to expand our knowledge of the geography of rape, information that is not readily available to researchers, given the sensitive nature of the data. Both law enforcement and research stand to benefit from strengthened collaborations, the sharing of data, and multidisciplinary learning. Investigators and crime analysts within police departments might have access to these data but rarely disseminate them after the offender has been caught. These SAK initiatives have allowed us to combine data points to examine a prolific rapist that greatly varied his MO over time.

For example, we explored the investigator’s theory about a connection between strip clubs and his selection sites. While many of the sites were near strip clubs, it is not possible from the data we have to know whether the relationship is directly (e.g., “hunting” for potential victims after leaving the strip club), indirectly (e.g., same neighborhoods that have strip clubs are also more likely to have residents who are walking and waiting), or spuriously related. Prior research suggests that these might at least be indirectly related [40,44,49], but more research using different data and methodologies would be needed to explore this connection in greater detail. However, by examining the spatial patterns for this large rape series, this study aligns with prior research on the relationship between outdoor rapes and alcohol-serving establishments [40,44,49], bus stops [49], and residences with higher crime [44].

### 6.1. Limitations

The data presented here are based on initiatives to address untested kits in two urban jurisdictions. They are not representative of all sexual assaults in this jurisdiction nor all reported sexual assaults with untested kits. Additionally, our data are derived from official police and prosecutor documentation, which varies by the officer, may contain errors, may fail to include essential details, and is based on information provided indirectly by the victims. Given that these data are from rapes that he was linked to through DNA from victims who reported and had kits collected, that he was a suspect in several others without DNA, and he appeared to become more forensically aware over time, there is a strong possibility that there are more rapes that have yet to be tied to him, which might provide a different insight into how and why he used the built environment in the commission of the crimes. As detailed in this study, little is known about the behavioral and spatial patterns of prolific stranger rapists. This study attempts to fill this knowledge gap via a case study of one prolific stranger rapist. In terms of future directions, more research should explore these more prolific stranger rapes [10] to assess how well our assumptions of site selection and behavior consistency hold up.

### 6.2. Implications for Practice

The findings speak to the value of all jurisdictions across the U.S. testing all of their kits and not just those where the suspect is a stranger and the victim wants to prosecute. As evidenced in this case study, rapists cross jurisdictional lines—if Akron had not also tested their kits, at least three fewer rapes would have been connected to him. Data from untested kits provide a more complete (but certainly not perfect) understanding of serial sexual assault offending.

Additionally, these findings have implications for how law enforcement understands sexual assault offenders’ site selection and behavioral decision-making processes. Serial stranger rapists might not necessarily maintain a consistent MO or victim type. While this offender was only connected to stranger rapes, he exhibited a great deal of crossover in victim type and variation in MO across offenses. The findings suggest that when attempting to link rapes without the use of DNA, law enforcement should consider casting a broader behavioral net. Moreover, offenders’ repeated use of certain locations (regardless of whether the MO or victim type is the same) might provide useful investigative leads.

Lastly, in the context of this special issue on health geography for effective interventions, the findings presented have implications for several fields, including victimology, law enforcement, spatial epidemiology, and public health for advancing our knowledge of the spatial patterns of prolific stranger rapists, the impact that these crimes have on communities that are repeatedly targeted, and more effective methods for interventions and prevention.

This study speaks to the lasting effects that these types of repeatedly committed crimes have on the public health of the residents who live and commute in those areas. Extant research suggests that children who grow up in communities rife with crime and violence are profoundly harmed, experiencing long-term mental and physical effects [51]. Additionally, this evidence suggests that neighborhood violence impacts the long-term health of its members and the economic development of the entire community, specifically within communities of color [51]. Relying solely on the criminal justice system to protect communities that experience repeated and targeted violence is fragmented and ineffective. Public health approaches that seek to prevent this type of violence and this type of criminal from being able to target these vulnerable victims and neighborhoods must be implemented.

In court transcripts for one of the trials for this offender, the judge overseeing the case stated that “found by clear and convincing evidence that defendant randomly selected his victims and had no prior relationship with any of them.” However, as we have shown, violence is not as truly ‘random’ as it appears at first glance. This offender exploited victims’ vulnerabilities—whether long-term (sex workers), temporary (home alone), or in situ (wrong place at the wrong time). Instead, violence is an epidemic that spreads, clusters, and transmits through communities once it begins. With a better understanding of how serial offenders take advantage of vulnerable communities, utilize space, and target their victims, we hope to continue to work towards preventing this type of violence and protecting potential victims from harm.

## Figures and Tables

**Figure 1 ijerph-19-06810-f001:**
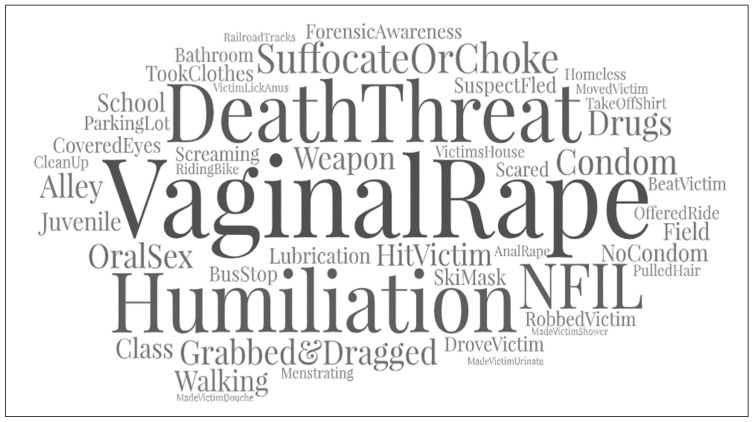
Word Cloud of the Police Reports for Offender’s 22 Rapes.

**Figure 2 ijerph-19-06810-f002:**
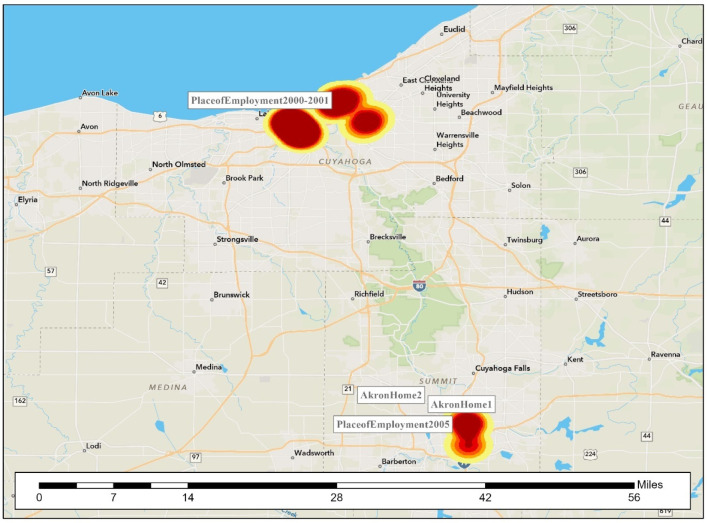
Offender’s Awareness Space, Sexual Assault Heat Map.

**Figure 3 ijerph-19-06810-f003:**
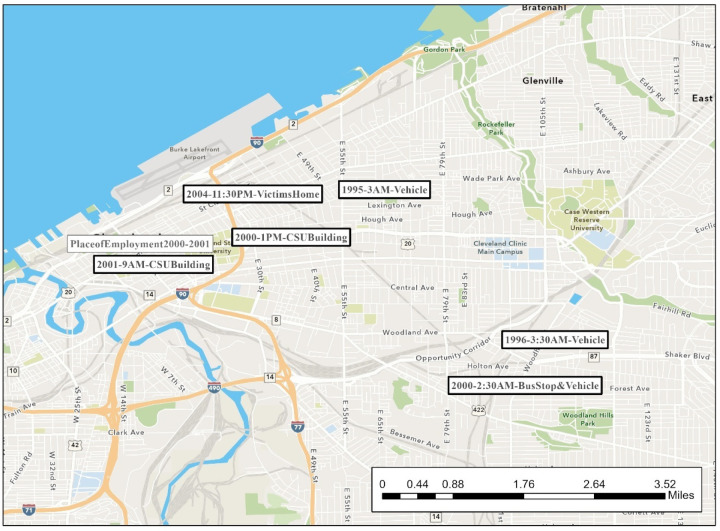
Cleveland’s East Side Sexual Assault Locations.

**Figure 4 ijerph-19-06810-f004:**
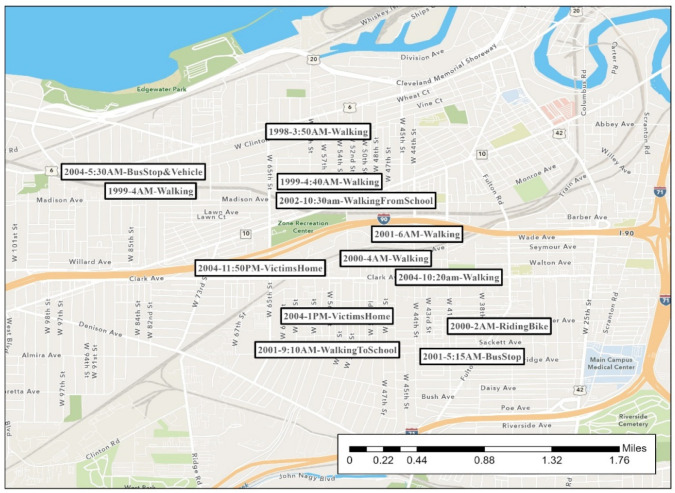
Cleveland’s West Side Sexual Assault Locations.

**Figure 5 ijerph-19-06810-f005:**
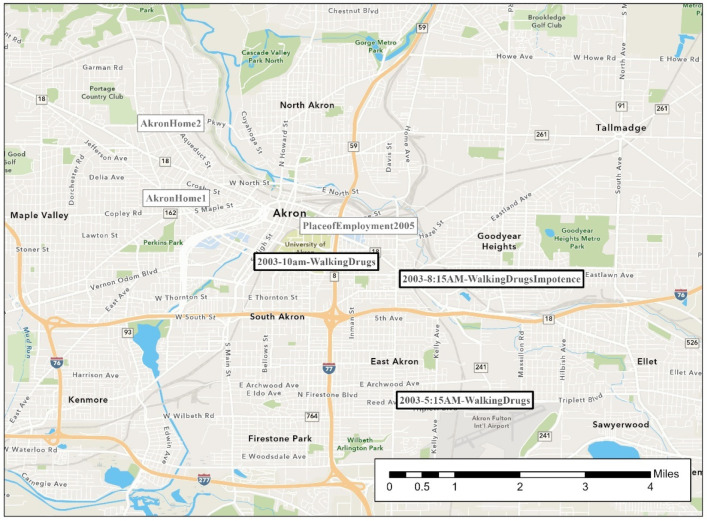
Akron’s Sexual Assault Locations, Offender Place of Employment, and Offender Known Residence(s).

**Figure 6 ijerph-19-06810-f006:**
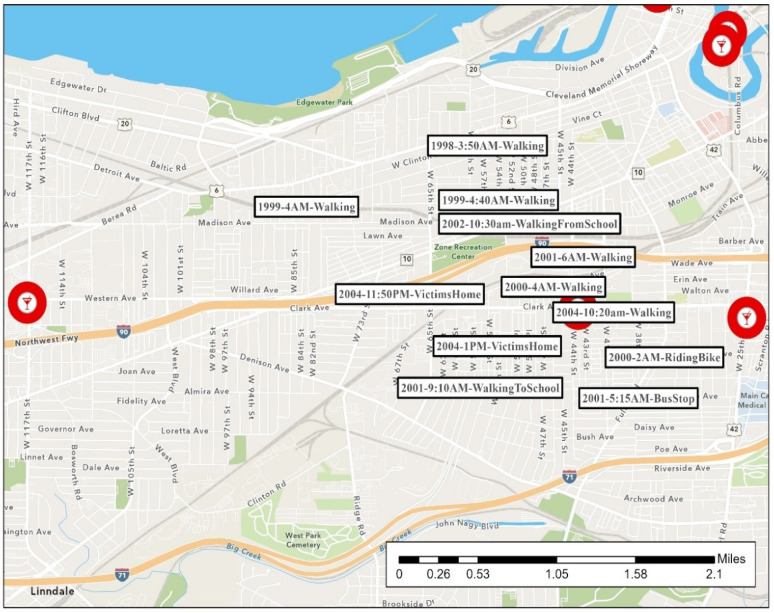
Strip Club Theory Map, Cleveland’s West Side (Strip Clubs Denoted With Red Martini Glass Symbols).

**Figure 7 ijerph-19-06810-f007:**
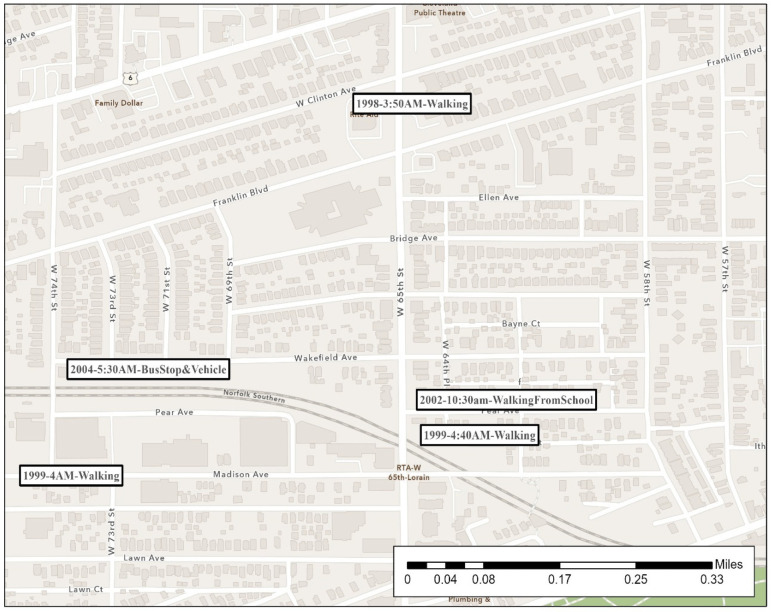
Cleveland’s West Side Favored Sexual Assault Locations.

**Table 1 ijerph-19-06810-t001:** Summary of the Offender’s Criminal History by Year and City.

Year	Charge
1988	Convicted, misdemeanor, disorderly conduct in Akron (Summit County)
Early 2001	Convicted, misdemeanor, soliciting in Akron (Summit County)
Spring 2001	Arrested (later convicted in 2001), misdemeanor telephone harassment and aggravated menacing of his ex-girlfriend, Painesville (Lake County)
Late 2001	Arrested (later convicted in 2002), misdemeanor indecent exposure in Erie, Pennsylvania (125 miles northeast of Akron). The case was sealed, so we were unable to obtain access to the details of this sexually based crime.

**Table 2 ijerph-19-06810-t002:** Overview of offender’s 22 sexual assaults.

Year	Victim Race	Time of Day	Means of Access	Location of Assault	Side of City	Victim Moved	Violence Used and Forensic Awareness
1995	Black	3:00 a.m.	Grabbed While Walking	In Suspect’s Car	East Cleveland	Transported In Car	Grabbed, Dragged, Gun, Robbed—No Condom
1996	Black	3:30 a.m.	Coerced Into Car	In Suspect’s Car	East Cleveland	Transported In Car	Hit, Gun, Humiliation, Covered Mouth—Condom
1998	White	3:50 a.m.	Grabbed While Walking	Behind A Store	West Cleveland	Dragged By Throat	Choked, Punched, Took Shirt—No Condom
1999	Black	4:00 a.m.	Grabbed While Walking	Empty Playground	West Cleveland	Dragged By Throat	Choked, Dragged, Knife—No Condom
-	White	4:40 a.m.	Grabbed While Walking	Railroad Tracks	West Cleveland	Dragged Down Steps To Railroad	Thrown Down, Choked, Beat—Condom
2000	Asian	1:00 p.m.	Followed Into Bathroom At School	Campus Building	Downtown	Moved Into Bathroom Stall	Hit In The Head, Death Threat—Condom
-	White	4:00 a.m.	Grabbed While Walking	Behind A School	West Cleveland	Dragged By Throat	Thrown Drown, Threatened, Took Clothing—No Condom
-	White	2:00 a.m.	Grabbed While Riding Bike	Rear Of Alley	West Cleveland	Dragged By Hair	Punched, Dragged, Death Threat, Choked, Smothered—No Condom
-	White	2:30 a.m.	Offered Ride While Waiting For Bus	Empty Field	West to East Cleveland	Transported In Car From West To East Cleveland	Choked, Dragged, Threatened — No Condom
2001	Black	5:15 a.m.	Grabbed Waiting For Bus	Behind An Alley	West Cleveland	Forced To Move From Bus Stop To Alley	Discussion Of Being Shot—No Condom
-	White	9:00 a.m.	Followed Into Classroom	Campus Building	Downtown	No Movement	Brutally Beaten—Unknown Condom
-	White	9:10 a.m.	Grabbed While Walking To School	Behind A Garage	West Cleveland	Dragged By Throat	Dragged, Choked, Death Threat, Knife, Took Clothing—No Condom
-	White	6:00 a.m.	Grabbed While Walking	Rear Of Building Under A Tree	West Cleveland	Forced By Gun To Rear Of Building	Death Threat, Gun, Took Clothing—Condom and Covered Eyes
2002	White	10:30 a.m.	Grabbed While Leaving School	Railroad Tracks	West Cleveland	Dragged Down Steps To Railroad	Death Threat, Punched, Dragged, Humiliation, Sadistic—Condom and Forced Victim To Urinate After
2003	White	5:15 a.m.	Grabbed While Walking (Victim On Drugs)	Field Behind Bar	Akron	Dragged Behind Building	Death Threat, Punched, Gun Dragged—Unknown Condom and Covered Eyes
-	White	8:15 a.m.	Grabbed While Walking (Victim On Drugs)	Behind A Garage	Akron	Dragged Behind A Garage	Dragged, Punched, Choked, Robbed—Condom and Covered Eyes
-	White	10:00 a.m.	Grabbed While Walking (Victim On Drugs)	Abandoned Building	Akron	Dragged To An Abandoned Building	Death Threat, Dragged, Choked—Condom and Covered Eyes
2004	White	1:00 p.m.	Broke Into Victim’s Home	Victim’s Bedroom	West Cleveland	Dragged Around Victim’s Home	Death Threat, Knife, Choked, Dragged, Robbed, Sadistic—Condom, Made Victim Lay On Towel, Took Her Phone, Made Her Shower and Rinse Mouth Out
-	White	11:50 p.m.	Broke Into Victim’s Home	Victim’s Home	West Cleveland	Dragged Around Victim’s Home	Death Threat, Gun, Dragged By Hair, Sadistic—Condom, Wore Ski Mask
-	White	11:30 p.m.	Broke Into Victim’s Home	Victim’s Bedroom	East Cleveland	Dragged Around Victim’s Home	Death Threat, Robbed, Choked, Sadistic—Condom, Wore Ski Mask/Gloves, Made Victim Lay On Towels, Took Towels, Made Victim Shower and Douche
-	White	10:20 a.m.	Grabbed While Walking	Basement Of A Home	West Cleveland	Transported In Car	Dragged By Hair, Pushed, Gun, Death Threat—Condom
-	White	5:30 a.m.	Grabbed Waiting For Bus	Railroad Tracks	West Cleveland	Transported In Car	Knife, Dragged, Punched, Choked, Sadistic—Condom, Told Victim He Was Taking Condom To Not Leave Evidence Behind

## Data Availability

Not applicable.

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
