# Peer review of "Understanding the Geography of Rape through the Integration of Data: Case Study of a Prolific, Mobile Serial Stranger Rapist Identified through Rape Kits"

_ijerph, 2022, doi:10.3390/ijerph19116810_

Round 1

Reviewer 1 Report

The manuscript is very well prepared. I have just a few recommendations: 

1. there is no clear description for what reason Figure 1 is done adn where is the description of it.

2. please provide a scale bars to the maps and legend to Figure 3.

3. could you please add the age of offenders in the Table 2?

4. in discussion section could you please describe the theoretical and methodical conclusions in separate paragraphs?

5. in conclusions section please answer to your research questions.

Author Response

Reviewer 1

Comment R1.1: There is no clear description for what reason figure 1 is done and where is the description of it.

Response: R1.1: We have included a statement in the below section (bolded below) that further clarifies why these analyses were conducted and included. 

(Pg 8; lines 367-369) “To examine these sexual assaults, Figure 1 shows a word cloud (also known as a tag cloud) of qualitative thematic codes of the 22 rapes. We qualitatively coded the narratives of the incident report to capture aspects of the rapes that were not collected as part of the close-ended quantitative variables and for validation of the quantitative findings. This step was also taken to get an initial understanding of the shared themes and patterns within the offender’s 22 assaults as his crimes were most often described as ‘random’ by law enforcement. Thematic coding was performed by hand in ATLAS.ti (ver. 9) and imported into a free word cloud software that allows for visual customization of the thematic coding. Word clouds are a straightforward way to represent textual data visually. According to researchers, these visual designs highlight more frequently used words, codes, or themes by allowing them to occupy more prominence (i.e., size) in the final representation [52]. Word clouds can be used for the preliminary analysis of text and validation of findings [52]. The larger or more prominent the coded word(s), the more frequently it occurred across the 22 rapes.”

We also added a description of it on page 8; lines 387-391.

Comment R1.2: Please provide a scale bar to the maps and legend to figure 3.

Response: R1.1: All maps, including Figure 3 have been reconfigured into full-color versions, now, they each include a greater degree of detail, a scale bar, and a description below.

Comment R1.3: Could you please add the age of offenders in table 2?

Response: R1.3: There is only one offender. Given the already large number of columns in Table 2, we have excluded the age of the offender. However, the details of his age over the series are on page 9, line 402. We state he was 28 in 1995 and his last known rape in this series was in late 2004. The year each rape occurred is provided in Table 2. Therefore, readers can easily calculate this age across the series. 

Comment R1.4: In the discussion section could you please describe the theoretical and methodical conclusions in separate paragraphs?

Response: R1.4: On page 17, starting at line 773, we have reorganized the discussion and included section headings for the theoretical and methodical discussion points.

Comment R1.5: In the conclusions section please answer your research questions.

Response: R1.5: We have reorganized the conclusion around the research questions and referenced the research questions to help tie the RQs back into the conclusion. (Starting pg 19; line 823).

Reviewer 2 Report

The paper is a very well written, high-quality publication.

The paper focuses on the consistency of the rapist's geographic site selection. The research proved the importance of SAK initiatives in unravelling stranger rapist cases. 

The paper first establishes the theoretical background behind their investigation. Then, point by point demonstrates the intersection of theory and practice in relation to a particular stranger rapist. This stranger rapist is highly mobile and not consistent in victim selection. Although in his site selection he was more persistent.

Although the paper quality is high I should mention the poor quality of the figures.  They are not helping the reader to understand the spatial aspects of the research, although it seems to be important.

The black background makes it difficult to read the figures. It would be nice to show on one map Akron, and the west and east sides of Cleveland. 

Because of the offender's journey to crime please put a scale bar on each map.

It is not clear what figure 2 is showing compared to Figures 3 and 4.

Figure 6, maybe put his cases onto the map also.

Author Response

Reviewer 2

Comment R2.1: Although the paper quality is high I should mention the poor quality of the figures. They are not helping the reader to understand the spatial aspects of the research, although it seems to be important. The black background makes it difficult to read the figures. It would be nice to show on one map Akron, and the west and east sides of Cleveland. Because of the offender's journey to crime please put a scale bar on each map. It is not clear what figure 2 is showing compared to figures 3 and 4. Figure 6, maybe put his cases onto the map also.

Response: R2.1: All figures have been reconfigured and updated in terms of quality. Each has been rendered in full color and has a scale included on each in order to better understand his journey to crime patterns. Figure 2 shows an overview of Akron and both sides of Cleveland but in terms of a heat map to better show which parts of the state he was favoring. Figure 6 has each of his cases on the map.

Reviewer 3 Report

Trying to use such sensitive data to reveal for us the patterns of rape by strangers is what makes this study especially laudable. However, after reading through, this article still contains some problems, as follows:

1. On the content of the article:

(1) The persuasiveness of the research results is weak.

First of all, the time range of the data involved in the study is from 1993 to 2011 and from 1993 to 2018. At this stage, social status and environmental characteristics have changed significantly with the development of cognition and technology. Whether the outdated data still has reference significance for the current management and control methods.

Second, the study involved only one repeat offender, are the serial perpetrator's behavioral data and behavioral preferences representative of this group? It is possible that other personnel are involved in fewer cases, such as 15 or 13 cases, but an appropriate increase in the number of subjects should be more convincing for the conclusion.

(2) In the third part "Materials and Methods", it is better to explain how to set the qualitative thematic codes of the 22 rapes.

(3) The practical significance of this study is ambiguous.

Although the data used in the full text are relatively extensive, they are actually relatively rough, which is of course a fact that everyone can understand. But after reading the full text, in addition to seeing the importance of DNA testing for the chain of cases, what are the specific recommendations and implications for the management and prediction of victims, law enforcement, space epidemics, and public health?

2. In terms of presentation form:

pictures 2, 6, and 7 are not clear, so the content cannot be displayed well

Author Response

Reviewer 3

Comment R3.1: The persuasiveness of the research results is weak. First of all, the time range of the data involved in the study is from 1993 to 2011 and from 1993 to 2018. At this stage, social status and environmental characteristics have changed significantly with the development of cognition and technology. Whether the outdated data still has reference significance for the current management and control methods.

Response: R3.1: These are the time points for the entire initiative in these jurisdictions. The first rape in the series is 1995 and the last is 2004. The kits were not all tested contemporaneously to the rapes. The first 9 kits were tested around the mid-to-late 2000s by Cleveland Police. The other kits were tested as part of the untested sexual assault kit initiatives in Cuyahoga and Akron in the early-to-mid 2010s. All were tested by the same crime lab. So, the technological advancements did not change that much. This is referenced on page 10; lines 441-445.

Comment R3.2: Second, the study involved only one repeat offender, are the serial perpetrator's behavioral data and behavioral preferences representative of this group? It is possible that other personnel are involved in fewer cases, such as 15 or 13 cases, but an appropriate increase in the number of subjects should be more convincing for the conclusion.

Response: R3.2: We argue here that understanding the behavior of a prolific serial perpetrator’s movements, patterns, and ultimately demise could give us insight as to how certain neighborhoods and populations of victims are often overlooked by law enforcement and targeted by these types of criminals. Why was this offender able to target the same neighborhoods year after year without the community catching onto his predatory habits? How was he able to assault two students on a college campus within almost a year of each other and still not get caught? The value of a case study analysis particular to criminology and public health is that it provides us with an opportunity to gain a greater understanding of serial rapists while at the same time reducing the potential for bias by diluting the agenda done to one particular individual.

We have included a statement in the limitations section that specifically states that we are using a case study, so we are unable to generalize these conclusions to all prolific stranger rapists (pg 20; line 920-924).

Comment R3.3: In the third part "materials and methods", it is better to explain how to set the qualitative thematic codes of the 22 rapes.

Response: R3.3: Given Review 1’s similar comment, the below statement (bolded below) that has now been included in the manuscript should help clarify the coding.

(Pg 8; lines 367-369) “To examine these sexual assaults, Figure 1 shows a word cloud (also known as a tag cloud) of qualitative thematic codes of the 22 rapes. We qualitatively coded the narratives of the incident report to capture aspects of the rapes that were not collected as part of the close-ended quantitative variables and for validation of the quantitative findings. This step was also taken to get an initial understanding of the shared themes and patterns within the offender’s 22 assaults as his crimes were most often described as ‘random’ by law enforcement. Thematic coding was performed by hand in ATLAS.ti (ver. 9) and imported into a free word cloud software that allows for visual customization of the thematic coding. Word clouds are a straightforward way to represent textual data visually. According to researchers, these visual designs highlight more frequently used words, codes, or themes by allowing them to occupy more prominence (i.e., size) in the final representation [52]. Word clouds can be used for the preliminary analysis of text and validation of findings [52]. The larger or more prominent the coded word(s), the more frequently it occurred across the 22 rapes.”

We also added a description of it on page 8; lines 387-391.

Comment R3.4: The practical significance of this study is ambiguous. Although the data used in the full text are relatively extensive, they are actually relatively rough, which is of course a fact that everyone can understand. But after reading the full text, in addition to seeing the importance of DNA testing for the chain of cases, what are the specific recommendations and implications for the management and prediction of victims, law enforcement, space epidemics, and public health?

Response: R3.4: We have added a section heading and reorganized the conclusion so as to highlights the practical applications of these findings. This revision starts on page 21; line 926.

Comment R3.5: In terms of presentation form: pictures 2, 6, and 7 are not clear, so the content cannot be displayed well

Response: R3.5: All figures, including 2, 6, and 7 have been reconfigured and updated in terms of quality. Each has been rendered in full color and has a scale included on each in order to better understand the offender’s journey to crime patterns.